# Regulation and Therapeutic Targeting of MTHFD2 and EZH2 in KRAS-Mutated Human Pulmonary Adenocarcinoma

**DOI:** 10.3390/metabo12070652

**Published:** 2022-07-15

**Authors:** Yuchan Li, Omar Elakad, Sha Yao, Alexander von Hammerstein-Equord, Marc Hinterthaner, Bernhard C. Danner, Carmelo Ferrai, Philipp Ströbel, Stefan Küffer, Hanibal Bohnenberger

**Affiliations:** 1Institute of Pathology, University Medical Center, 37075 Göttingen, Germany; yuchan.li@stud.uni-goettingen.de (Y.L.); omarakkad14@gmail.com (O.E.); yaosha2016@gmail.com (S.Y.); carmelo.ferrai@med.uni-goettingen.de (C.F.); philipp.stroebel@med.uni-goettingen.de (P.S.); 2Department of Internal Medicine 2, Goethe University Hospital, 60590 Frankfurt, Germany; 3Department of Pathology, The 3rd Xiangya Hospital, Central South University, Changsha 410013, China; 4Department of Thoracic and Cardiovascular Surgery, University Medical Center, 37075 Göttingen, Germany; alexander.hammerstein@med.uni-goettingen.de (A.v.H.-E.); marc.hinterthaner@med.uni-goettingen.de (M.H.); bernd.danner@med.uni-goettingen.de (B.C.D.)

**Keywords:** one-carbon metabolism, MTHFD2, KRAS, EZH2, pulmonary adenocarcinoma

## Abstract

Activating KRAS mutations occur in about 30% of pulmonary adenocarcinoma (AC) cases and the discovery of specific inhibitors of G12C-mutated KRAS has considerably improved the prognosis for a subgroup of about 14% of non-small cell lung cancer (NSCLC) patients. However, even in patients with a KRAS G12C mutation, the overall response rate only reaches about 40% and mutations other than G12C still cannot be targeted. Despite the fact that one-carbon metabolism (1CM) and epigenetic regulation are known to be dysregulated by aberrant KRAS activity, we still lack evidence that co-treatment with drugs that regulate these factors might ameliorate response rates and patient prognosis. In this study, we show a direct dependency of Methylenetetrahydrofolate dehydrogenase 2 (MTHFD2) and Enhancer of Zeste Homolog 2 (EZH2) expression on mutationally activated KRAS and their prognostic relevance in KRAS-mutated AC. We show that aberrant KRAS activity generates a vulnerability of AC cancer cell lines to both MTHFD2 and EZH2 inhibitors. Importantly, co-inhibition of both factors was synergistically effective and comparable to KRASG12C inhibition alone, paving the way for their use in a therapeutic approach for NSCLC cancer patients.

## 1. Introduction

Lung cancer is one of the most aggressive and deadliest cancer types [1] and causes about 1.4 × 10^6^ global deaths every year [2]. Pulmonary adenocarcinoma (AC) is the main subgroup of non-small cell lung cancer, accounting for nearly 40% of all cases [2,3]. On the molecular level, AC can be further subdivided, and activating mutations of KRAS define the largest molecular subgroup (~30%) [4,5]. For a long time, KRAS mutations were considered to be undruggable, but, eventually, the KRAS G12C (KRAS^G12C^) specific inhibitor AMG510 (Sotorasib) was FDA-approved for second-line treatment of AC patients. However, mutations other than G12C are still not targetable and the overall response rate for Sotorasib among KRAS^G12C^ cases remains below its expected level, with about 60% of patients evading response [6,7,8,9].

KRAS mutations play a vital role in controlling tumor metabolism, e.g., by stimulating glucose uptake [10,11,12], and an increased need for energy and elevated aerobic glycolysis are closely associated with chemoresistance, tumor progression, and metastasis of malignant tumors [13,14,15,16]. We and others have recently shown that KRAS mutations are connected to an enhanced dependency on one-carbon metabolism (1CM) in non-small cell lung cancer (NSCLC) and colorectal cancer [17,18,19]. One-carbon metabolism includes the methionine and folate cycles and is essential for the maintenance of cellular homeostasis. Integrating the cell’s nutritional status 1CM catabolizes different carbon sources to derive one-carbon (methyl) units. In cancer cells, the high proliferation rate requires these one-carbon units for nucleotide synthesis, methylation, and reductive metabolism [20]. Furthermore, we have shown that Methylenetetrahydrofolate dehydrogenase 2 (MTHFD2) is essential for the survival of NSCLC cell lines and is a prognostic factor in AC [18]. MTHFD2 is one of the key enzymes in 1CM and is strongly expressed in embryonic development but it is almost absent in most healthy adult tissues, making it a promising potential therapeutic target for cancer treatment [21]. High levels of MTHFD2 have been associated with tumor recurrence and worse prognosis in multiple solid and liquid malignancies and participate in resistance against gemcitabine and pemetrexed [18,19,21,22,23,24,25,26,27,28].

Enhancer of Zeste Homolog 2 (EZH2) is a member of the polycomb repressive complex 2 (PRC2), which plays an important role in maintaining cellular identity by regulating the transcription of genes through deposition of the H3K27me3 repressive mark. EZH2 is upregulated in multiple malignancies, including NSCLC [29,30,31,32,33,34,35]; therefore, several attempts have been made to inhibit EZH2 as a clinical treatment approach. The specific EZH2 inhibitors GSK126 and EPZ-6438 have yet not reached clinical stages [36,37]; however, tazemetostat was granted accelerated approval at the beginning of 2020 by USFDA for Follicular Lymphoma and has since been tested in multiple solid tumors [38,39,40]. Recent findings connected EZH2 expression to KRAS mutations and metabolism. Compared to a moderate expression in KRAS wild-type (KRAS^WT^) cell lines, expression of EZH2 is increased in cells with an activating KRAS mutation [35]. EZH2 has also been described to facilitate metabolic reprogramming in glioblastomas with a substantial increase in glycolytic metabolism [41].

A better understanding of the metabolic and epigenetic network regulated by aberrant KRAS in AC is therefore of primary importance. In the present study, we investigated the expression of MTHFD2 and EZH2 as dependent on KRAS mutational status in a cohort of primary AC patient samples. Our results highlight a functional connection between mutated KRAS, EZH2, and MTHFD2 and reveal a vulnerability of KRAS-mutated AC cancer cell lines to both MTHFD2 and EZH2 inhibitors. Importantly, our study shows that co-inhibition of both factors was synergistically effective in addition to KRAS^G12C^ inhibition alone, providing evidence that their use in a therapeutic approach for NSCLC cancer patients may increase overall response rate outcomes in KRAS^G12C^ cases.

## 2. Results

### 2.1. EZH2 and MTHFD2 Expression Correlate with KRAS Mutation Status and Clinicopathologic Characteristics in AC Patients

KRAS mutations have been connected to metabolism and epigenetic regulation in several cancer types [35,42,43]. Therefore, we investigated a cohort of 109 AC patients for activating KRAS mutations and protein expression of MTHFD2 and EZH2. The clinical characteristics of the patients are summarized in (Table 1 and Appendix A). All patients underwent surgical tumor resection without prior chemotherapy. Male patients (56%) were slightly more numerous than females and the median age at the time of diagnosis was 67 years (range 34–85 years). Most patients demonstrated a moderately differentiated disease, the frequency of T1–2 stage was 82.6%, and the majority were lymph node-negative (60.2%). The median follow-up time was 23 months and 48% of the patients deceased during the follow-up time.

Among 62 patients with a clinical follow-up, we examined the mutational status of exon 2 of the KRAS gene. Eighteen samples were KRAS-mutated (G12C n = 15, G12V n = 3) and 44 were found to be KRAS^WT^. The MTHFD2 statuses of the investigated samples were published previously [18] and can be found in Appendix A. Comparing MTHFD2 expression in KRAS^MUT^ and KRAS^WT^ samples, we observed a significant correlation between high expression of MTHFD2 and KRAS^MUT^ tumors (*p* = 0.0066) (Figure 1a). Furthermore, overall survival (OS) was significantly influenced by MTHFD2 expression in KRAS^MUT^ (*p* = 0.0178, Figure 1b) but not KRAS^WT^ cases (*p* = 0.2906, Figure 1c).

EZH2 was strongly expressed in 38.5% of investigated cases and high expression indicated a significantly shorter OS compared to low EZH2 levels (*p* = 0.0027) (Figure 1d–f). As for MTHFD2, EZH2 high expression was more common in KRAS^MUT^ cases (*p* = 0.0039, Figure 1g), and patient prognosis was significantly decreased when EZH2 was highly expressed in KRAS^MUT^ patients (*p* = 0.0419, Figure 1h) but not in KRAS^WT^ cases (*p* = 0.1126, Figure 1i).

Next, we have shown that strong expression of MTHFD2 was significantly associated both with KRAS mutations and high levels of EZH2 (*p* = 0.0005, Figure 1j). Patients with strong expression of both EZH2 and MTHFD2 harboring a KRAS mutation had the worst prognosis (*p* = 0.0128, Figure 1k), whereas no influence on OS was detected based on protein expression of EZH2 and MTHFD2 in KRAS^WT^ and (*p* = 0.1152, Figure 1l).

In addition, EZH2 expression and KRAS mutation significantly correlated with the occurrence of lymph node metastasis (*p* = 0.0003 and *p* = 0.0052, respectively) and poor differentiation grade (*p* < 0.0001 and *p* = 0.035) as shown in Table 2.

### 2.2. Expression of MTHFD2 and EZH2 Depends on the Activity of Mutated KRAS in Human Pulmonary Adenocarcinoma Cell Lines

To further investigate the functional interplay between MTHFD2, EZH2, and mutated KRAS, we used two KRAS^G12C^-mutated (HCC44 and H23) and two KRAS^WT^ (H1993 and HCC78) AC cell culture models. Western blot analyses revealed up to 70% increased levels of MTHFD2 and EZH2 in HCC44 and H23 in comparison to H1993 and HCC78 (Figure 2a and Appendix A). Treating the cells with 4μM of the KRAS^G12C^-specific inhibitor AMG510 for 48h showed a significant decrease in cellular survival only in the KRAS-mutated cell lines HCC44 and H23 (Figure 2b). Furthermore, EZH2 and MTHFD2 protein levels were decreased only in HCC44 and H23 after AMG510 treatment, whereas there were no changes in H1993 and HCC78 (Figure 2c). Transient overexpression of KRAS^G12C^ in H1993 (KRAS^C12Gvec^) also raised the expression of MTHFD2 and EZH2 by about 30% (Figure 2d and Appendix A). Treatment with AMG510 (4 μM, 48 h) reduced MTHFD2 and EZH2 expression only in H1993 (KRAS^C12Gvec^) (Figure 2e). Additionally, cellular survival significantly decreased only in KRAS^G12C^-overexpressing H1993 cells and not in cells containing the KRAS^WT^ vector (KRAS^WTvec^) (Figure 2f). Altogether, the results show that the aberrant activity of KRAS^G12C^ cells increases the expression of MTHFD2 and EZH2 in human pulmonary AC cell lines.

### 2.3. EZH2 Repressive Activity Is Required to Modulate MTHFD2 Expression in KRAS^G12C^ Cell Lines

To characterize the relationship between EZH2, MTHFD2, and KRAS in AC cells, we knocked down EZH2 using two specific siRNAs. Interestingly, next to EZH2, MTHFD2, also, was robustly downregulated only in the KRAS^G12C^ cell lines HCC44 and H23 (Figure 3a,c and Appendix A). Consistently, treatment with the EZH2 methyl transferase-specific inhibitor GSK126 showed a decrease in H3K27me3 in all cells, but a decreased expression of MTHFD2 was only observed in the KRAS^G12C^ cell lines (Figure 3b). We show, also, that cellular viability upon GSK126 treatment was significantly decreased in KRAS^G12C^ H1993 KRAS^G12Cvec^ cell lines compared to KRAS^WT^ cell lines (Figure 3d,e).

On the other hand, there was no relevant impact on EZH2 protein expression upon MTHFD2 knockdown in either KRAS^G12C^ or KRAS^WT^ cells (Figure 3f,h and Appendix A). Similarly, treating cells with the MTHFD2-specific inhibitor DS (DS18561882) did not affect EZH2 expression (Figure 3h), although DS treatment strongly reduced the cellular viability of HCC44, H23, and H1993 KRAS^G12Cvec^ cells compared to H1993, Hcc78, and KRAS^WTvec^ cells (Figure 3i,j). Altogether, we show that MTHFD2 specifically targets the KRAS^G12C^ background and that, in turn, MTHFD2 expression is subjected to KRAS^G12C^ and EZH2 activity.

### 2.4. Combinational Treatment of KRAS^G12C^ with EZH2 and MTHFD2 Inhibitors

Based on our results that show a strong dependency of MTHFD2 and EHZ2 on activating KRAS mutations, we decided to test whether a combinational treatment may have a synergistic effect on AC viability. We therefore treated HCC44 with increasing concentrations of AMG510 in combination with (I) GSK126 or (II) DS or (III) GSK126 together with DS. Compared to single treatments, all combinations showed an increased and synergistic reduction in cellular survival (Figure 4a–f). In particular, the combination of GSK126 and DS showed a good response that was comparable to KRAS^G12C^ inhibition with relatively low concentrations.

We have previously reported that high MTHFD2 expression induced resistance against Pemetrexed (PTX) [18] and that downregulation of KRAS^G12C^ enhances response to PTX [17]. To test whether PTX resistance in HCC44 can be recovered by KRAS^G12C^ inhibition, we incubated HCC44 and H1993 cells with either AMG510 (2 μM), PTX (20 μM), or in combination. After a single treatment, EZH2 and MTHFD2 protein levels were only partially reduced. However, the combination almost completely reduced EZH2 and MTHFD2 specifically in HCC44, meanwhile the expression in H1993 was almost unaffected (Figure 4g). Furthermore, the combination of PTX treatment with GSK126 (2.5 μM) and DS (10 μM) was about 10-fold, and the combination of PTX with AMG510 (2 μM) was 80-fold stronger than the single treatment in HCC44 (Figure 4h). On the contrary, in H1993 cells, no additional effect was observed (Figure 4i).

## 3. Discussion

Metastasis and recurrence after tumor resection and adjuvant therapy occur in almost 70% of pulmonary adenocarcinoma cases and keep the five-year survival rate at approximately 15% [44]. The discovery of the small-molecule inhibitor AMG510 directed against G12C-mutated KRAS significantly improved the prognosis of many affected AC patients; however, besides the mutations other than G12C that cannot be targeted, less than 40% of treated patients responded to therapy [6,7,8]. In this study, we show by immunohistochemical and in vitro approaches that KRAS^G12C^ upregulates the epigenetic regulator EZH2 and the 1CM metabolic enzyme MTHFD2, leading to molecular vulnerabilities for combinational treatment approaches.

In our study, we found that expression of both MTHFD2 and EZH2 are increased when KRAS acquires activating mutations in patients with AC. Importantly, only in KRAS-mutated cases, high expression of both MTHFD2 and EZH2 was predictive of inferior patient prognosis. Aberrant KRAS activity has been described previously as rewiring metabolism and epigenetic regulation [35,42,43]. Riquelme et al. showed that in KRAS^G12C^-mutant NSCLC, EZH2 expression was preferentially upregulated through the MEK–ERK signaling pathway [35], and a negative prognostic association of EZH2 overexpression has been proposed in several human malignancies, including NSCLC [29,30,31,32,33,34,35]. Indeed, our data not only show a correlation between KRAS^G12C^ and EZH2 expression, but the functional ablation of EZH2 methylase activity also clearly supports its role in AC cancer cell viability. Epigenetic repression of *EAF2-HIF1α* by EZH2 has been shown to foster metabolic reprogramming in glioblastoma and to promote aerobic glycolysis by upregulating HK2 in prostate cancer [41,45]. In addition, high 1CM activity has been linked to increased tumor aggressiveness and reduced prognosis in several cancer entities, and a dependency of MTHFD2 on KRAS and its prognostic impact was described in AC, colorectal, and pancreatic cancer [18,19,46].

As expected, pharmacological inhibition of KRAS decreased cellular viability and reduced protein levels of MTHFD2 and EZH2 only in KRAS^G12C^ cells. By over-expressing KRAS^G12C^ in the KRASWT cell line H1993 we could validate a direct influence of KRAS activity on metabolism and the regulation of MTHFD2 and EZH2 expression. Specific knockdown or pharmacological inhibition revealed a KRAS^G12C^-dependent MTHFD2 regulation by EZH2, whereas siRNAs against MTHFD2 had no effect on EZH2 expression. This suggests that KRAS^G12C^ activity is responsible for the upregulation of MTHFD2 subjected to the control of EZH2 methyltransferase activity. The fact that our results show MTHFD2 down-regulation in response to EZH2 inhibition indicates that such regulation most likely occurs indirectly. One obvious mechanistic explanation is that EZH2 may target an MTHFD2 transcriptional repressor. Another possibility to consider is that 3D genome organization analyses indicate that Polycomb-bound loci form insulated and self-interacting chromatin domains [47] and that the removal of EZH2 activity may induce rewiring of the MTHFD2 gene locus, leading to the preclusion of interaction with its regulatory sequences. Polycomb marks are highly enriched at CpG islands (CGIs), and H3K27me3 distribution is known to be anti-correlated with DNA methylation [48]; in this regard, EZH2-silenced foci near the MTHFD2 gene may acquire DNA methylation (epigenetic switching) [49], triggering the constitutive silencing of the entire locus in response to PRC repression inhibition.

However, the fact that MTHFD2 is transcriptionally regulated by epigenetic fluctuations does not represent the only hypothesis. MTHFD2 is relatively lowly expressed in normal tissues [47] and is upregulated in different cancers, including breast, colorectal, and hepatocellular cancers, where it plays a key role in remodeling the folate metabolism of tumor cells [48]. There have been indications that KRAS regulates MTHFD2 at a transcriptional level [49], but post-translational regulation also seems possible. It has been proposed that MTHFD2 activity and proteasomal degradation are regulated by acetylation [50,51]. Zhang et al. [51] described that the acetylation of MTHFD2 by SIRT4 leads to increased proteasomal degradation in breast cancer. They describe SIRT4 as a guardian of cell metabolism and a sensor of folate availability. Since KRAS and EZH2 are known to increase the metabolic state and the availability of folate, it could be suggested that MTHFD2 is regulated by proteasomal degradation dependent on the metabolic state of the cell. In our study, we aimed to target MTHFD and EZH2 as a therapeutic approach for NSCLC cancer patients and further studies will address the mechanism behind their action. Strikingly, the inhibition of EZH2 or MTHFD2 strongly reduces cellular viability only in KRAS^G12C^ cell lines. The increased response to inhibition of EZH2 and MTHFD2 of the KRAS^WT^ cell line H1993 overexpressing KRAS^G12C^ further suggested that KRAS activity is a main driver of the epigenetic regulation and increased 1CM activity in AC. These findings indicate that the inhibition of EZH2 and MTHFD2 will be mainly effective in KRAS-mutated AC. MTHFD2 and EZH2 inhibition have been proposed as therapeutic options in several solid and lymphatic cancers [19,36,37,38,39,40,50,51]. Importantly, here we show that the combined inhibition of mutated KRAS and either EZH2 or MTHFD2 exhibit a synergistic effect in KRAS^G12C^ cells, and, surprisingly, the co-inhibition of EZH2 and MTHFD2 had a similar synergistic effect to the combination with AMG510. These results strongly suggest that AC patients with a KRAS mutation other than KRAS^G12C^ might also profit from combined inhibition of MTHFD2 and EZH2.

We previously demonstrated that strong expression of MTHFD2, as seen in KRAS-mutated cells, leads to resistance of AC cells against treatment with PTX, which is commonly used as first-line therapy for AC patients [18]. While the combination of PTX with AMG510, GSK126, or DS had no additional effect in the KRAS^WT^ cell line H1993, we discovered a synergistic effect when we combined PTX with GSK126 or DS in the KRAS^G12C^ cell line HCC44. Since the combination of PTX with AMG510 led to a similar response to PTX in the KRAS^WT^ cell line, we also suggest PTX as a treatment option in AMG510-treated KRAS^G12C^ AC with a moderate response.

In summary, our findings indicate a causal connection between KRAS mutation status and the expression of the epigenetic regulator EZH2 and the 1CM marker MTHFD2. KRAS-mutated AC cells are vulnerable to inhibition of EZH2 and MTHFD2, and combinational treatment is as efficient as KRAS^G12C^ inhibition alone, revealing a potential treatment option for non-KRAS^G12C^-mutated AC. In addition, we suggest PTX as a treatment option in AMG510-treated KRAS^G12C^ AC with moderate response. These findings may lead to a better separation of AC patients and an improved response rate by combinational treatment strategies.

## 4. Materials and Methods

### 4.1. Human Tissue Samples

The NSCLC tissue samples used in this study were collected from the Department of Thoracic and Cardiovascular Surgery, University Medical Center Göttingen after surgical resections. Tissues were formalin-fixed in 4% buffered formaldehyde and paraffin-embedded for diagnostic purposes. The performed experiments were approved by the ethics committee of the University Medical Center Göttingen (#1-2-08, 24-4-20) and all procedures were performed in accordance with the seventh version of the Declaration of Helsinki [52].

### 4.2. Immunohistochemical Staining

Tissue samples were assembled in tissue microarrays and EZH2 was immunohistochemically stained as described previously [53,54,55]. Briefly, 2 µm tissue sections were deparaffinized, rehydrated, and subsequently incubated in EnVision Flex Target Retrieval Solution pH high (Dako/Agilent, Santa Clara, CA, USA), followed by incubation with primary antibody against EZH2 (Leica, Wetzlar, Germany, NCL-L-EZH2, 1:50), secondary antibody (EnVision Flex+, Dako), and DAB (Dako) and counterstaining with Hematoxylin. Staining was evaluated under light microscopy according to the signal intensity of the stained cells: zero = no staining; one = weak staining intensity; two = strong staining intensity.

### 4.3. DNA Isolation and KRAS Exon 2 Profiling

DNA was isolated from 10 μm FFPE tissue sections with the InnuPREP FFPE DNA Extraction Kit (Analytik Jena, Jena, Germany), according to the manufacturer’s instructions. Considering the specificity of Sotarasib towards the G12C mutation in clinical applications, we exclusively profiled exon 2 of KRAS. Primers were designed using Primer3Web (https://primer3.ut.ee, accessed on 10 July 2022) accessed on 11 Dezember 3021. Primers spanning the specific genomic region were chosen (G12C FOR: 5′-GGCCTGCTGAAAATGAC-3′ and G12C REV: 5′-TGTATCAAAGAATGGTCCTGCAC-3′). Two-hundred nanograms of quantified DNA (Nanodrop, Peqlab, Erlangen, Germany) was PCR-amplified using 2 × MyTaqTM HS Mix (PCRBIOSYSTEMS, London, UK) with indicated KRAS primers on a labcycler (Sensoquest, Göttingen, Germany). PCR products were purified and subjected to Sanger sequencing on a 3500 Genetic Analyzer (Applied Biosystems Inc., Waltham, MA, USA) using the Applied BiosystemsTM Sanger Sequencing Kit (Applied Biosystems Inc., Waltham, MA, USA). Sequences were analyzed by comparing the acquired sequences with KRASWT using Geneious 11.1.3 software (http://www.geneious.com, accessed on 10 July 2022).

### 4.4. Cell Culture

The human AC cell lines HCC44, H23, H1993, and HCC78 were purchased from the American Type Culture Collection (ATCC, Manassas, VA, USA). Cells were maintained in RPMI-1640 supplemented with 1% L-Glutamine, 1% Penicillin–Streptomycin, and 10% fetal bovine serum (Gibco / ThermoFisher, Waltham, MA, USA) at 37 ℃ in a 5% CO_2_ humidified environment. The medium was refreshed three times per week and cells were passaged at approximate confluency of 80%.

### 4.5. MTS and ATP Assay

Cell survival analysis was performed using the CellTiter-Glo assay and the MTS assay kit (Promega, Madison, WI, USA), according to the manufacturer’s protocol. The CellTiter-Glo assay was used to perform the ATP luminescence assay. Chemiluminescence was measured using a Tecan Reader Infinite 2000 (Tecan, Männedorf, Switzerland).

### 4.6. Cell Transfection with siRNA and Expression Plasmids

All siRNAs in this study were obtained from Qiagen (Qiagen, Hilden, Germany). For the cell transfection, 3 × 10^5^ cells were transfected with 30 nM siRNA against MTHFD2 (#1 SI02664928, #2 SI02664921) or 80 nM siRNA against EZH2 (#1 SI02633316, #2 SI02665166) using HiPerFect Transfection Reagent (Qiagen), according to the manufacturer’s protocol. Allstars negative siRNA was used as a scrambled control. Plasmid transfection was performed with XtremeGENE HP DNA transfection reagent (Merck, Darmstadt, Germany). In brief, a 100 µL transfection mix containing 4 μL transfection reagent, 2 μg expression vector DNA, and serum-free RPMI-1640 cell culture medium was incubated at room temperature for 15 min and added to 3 × 10^5^ cells seeded on a 6-well plate in 2 mL medium. Either a pCMV6-Entry-KRAS^G12C^ vector (Origene Technologies Inc., Rockville, MD, USA) or a pBabe-KRAS^WT^ vector (Addgene, Watertown, MA, USA) was transfected into H1993. Cells expressing de novo KRAS^G12C^ and KRAS^WT^ were selected with G418 and Puromycin at concentrations of 800 µg/mL and 2 µg/mL, respectively, for at least 10 days, and KRAS protein levels were confirmed by Western blotting.

### 4.7. Western Blotting

Cells were lysed with NP40 buffer at a pH of 7.6 containing 50 mM Tris, 150 mM NaCl, 1% NP40, 0.2% Lauryl Maltoside, 1 mM Sodium orthovanadate (Sigma-Aldrich, St. Louis, MO, USA), and 1 × protease inhibitor cocktail of cOmplete™ (Roche, Basel, Switzerland). Protein concentration was determined using a DC™ Protein Assay kit (Bio-Rad, Hercules, CA, USA), and proteins were immunoblotted as described previously [56]. Then, 15 µg denatured protein was separated on precast Mini Protein TGX gels (Bio-Rad) and transferred to a nitrocellulose membrane using the semi-dry Trans-Blot Turbo™ system. Antibodies and related secondary antibodies (Dako / Agilent, Santa Clara, CA, USA) were used at a dilution of 1:1,000 in TBST for Anti-EZH2 (Cell Signaling Technology, Danvers, MA, USA, #5246), Anti-H3K27me3 (ActiveMotif, Carlsbad, CA, USA, #39155), and Anti-MTHFD2 (Abnova, Heidelberg, Germany, #H00010797-M01). Anti-PARK7 (Abcam, Camebridge, UK, #ab18257) and Anti-GADPH (Cell Signaling Technology, #5174) were used as loading controls.

### 4.8. Drug Treatment Assays

Fifteen-hundred cells were seeded on a 96-well plate 24 h before drug treatment. The specific drugs used were AMG510 (MedChemExpress, USA, #HY-114277), GSK126 (Selleckchem, Houston, TX, USA, #S7061), and DS (DS18561882) (MedChemExpress, Monmouth Junction, NJ, USA, HY-130251). Cells were treated with indicated concentrations for indicated time periods. For synergistic effect assays, two inhibitors were combined in a series of nine increasing concentrations. Inhibitor concentrations ranged from 0 to 25 μM for AMG510, 0 to 20 μM for GSK126, and 0 to 100 μM for DS. In corresponding control groups, an equivalent amount of DMSO was added. After 72 h treatment, cell viability was analyzed using MTS, as described above. Drug synergy was evaluated using CompuSyn software (ComboSyn Inc, New York, NY, USA) for the combination index (CI) value calculations. CI values: antagonism (CI > 1.1), additive effect (CI < 1.1 and CI > 0.9), and synergism (CI < 0.9) [57].

### 4.9. Statistical Analysis

Statistical analysis was performed using GraphPad (GraphPad Software LCC, San Diego, CA, USA). Two-group comparisons were performed with Students’ *t*-tests. Half-maximal inhibitory concentration (IC50) analysis was performed using Pearson’s correlation test. Cell growth and resistance comparisons were analyzed using two-way ANOVA. Survival analyses were performed using the Kaplan–Meier method and the log-rank (Cox–Mantel) test. ImageJ was used for the quantification of WB signal intensities [58]. Three biological replicates were performed. The data depicted are the means ± SEMs. A *p*-value of <0.05 was considered significant (* *p* < 0.05, ** *p* < 0.01, *** *p* < 0.001).

## Figures and Tables

**Figure 1 metabolites-12-00652-f001:**
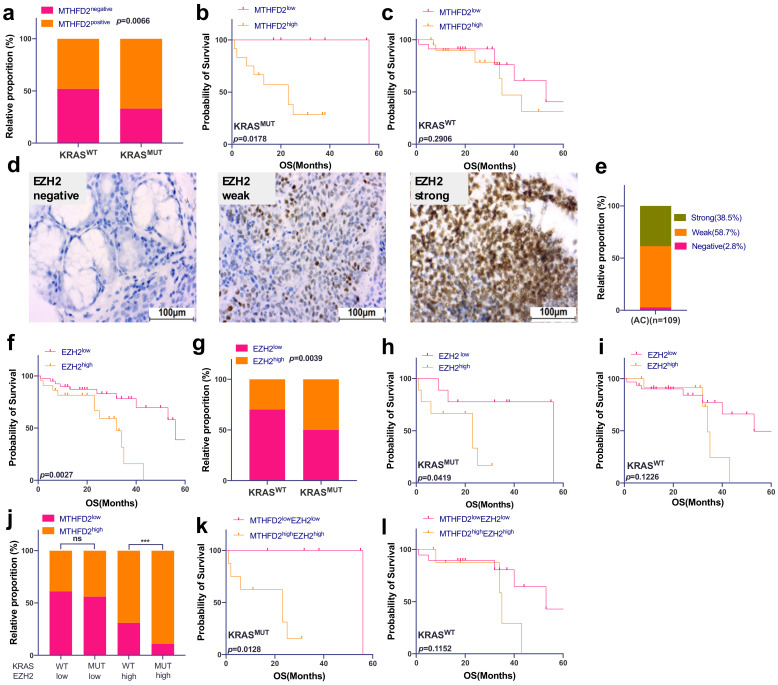
EZH2 and MTHFD2 expression correlate with KRAS mutation status and patient survival in AC patients. (**a**) Correlation between MTHFD2 protein expression and KRAS gene mutation in AC samples (*p* = 0.0066). (**b**) Kaplan–Meier survival analysis of AC according to MTHFD2 protein expression in KRAS^MUT^ samples (*p* = 0.0178) and (**c**) KRAS^WT^ samples (*p* = 0.2906). (**d**) Human AC tissues were immunohistochemically stained for EZH2. All images were captured at 40× magnification. (**e**) Prevalence of EZH2 protein expression in AC patients. (**f**) Kaplan–Meier survival analysis according to EZH2 protein expression in AC patients. (**g**) Correlation between EZH2 protein expression and KRAS gene mutation in AC samples (*p* = 0.0039). (**h**) Kaplan–Meier survival analysis of AC according to EZH2 expression in KRAS^MUT^ (*p* = 0.0419) and (**i**) KRAS^WT^ (*p* = 0.1226) patients. (**j**) Correlation between MTHFD2 protein expression grouped by EZH2 protein expression and KRAS mutational status. (**k**) Kaplan–Meier survival analysis of AC according to both MTHFD2 and EZH2 expression in KRAS^MUT^ (*p* = 0.0128) and (**l**) KRAS^WT^ (*p* = 0.1152) patients. (**d**,**e**) were performed for n = 109; all the other analyses were matched samples, n = 62.

**Figure 2 metabolites-12-00652-f002:**
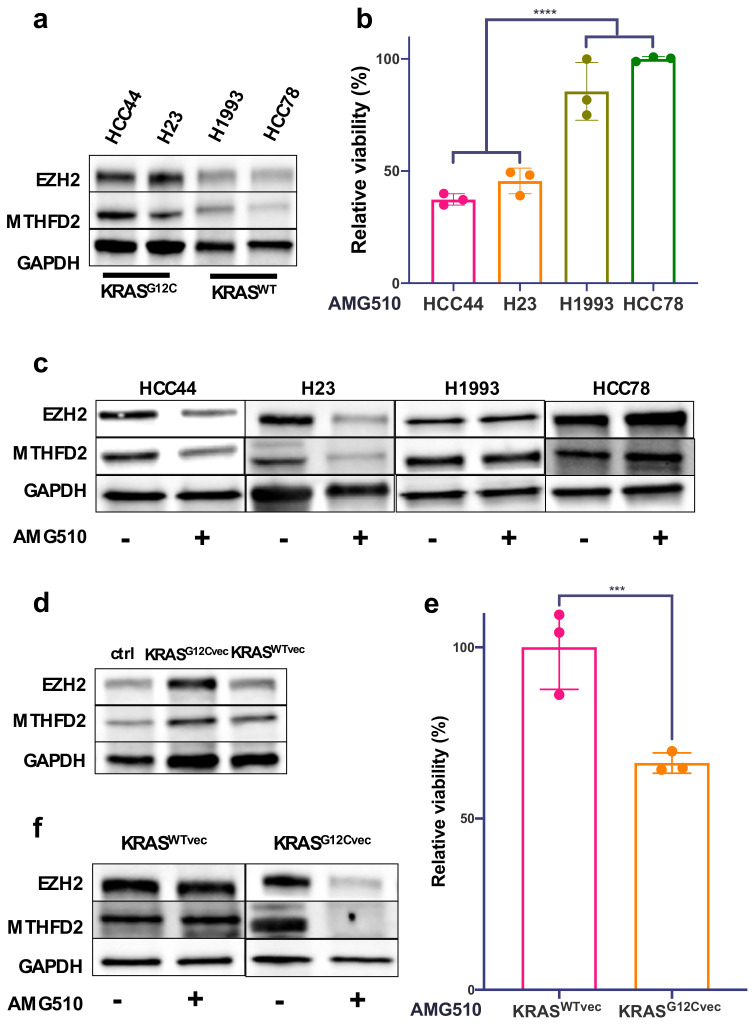
Expression of MTHFD2 and EZH2 depends on mutated KRAS in human AC cell lines. (**a**) Western blot analysis of EZH2 and MTHFD2 expression in the AC cell lines HCC44, H23, H1993, and HCC78. (**b**) Cellular survival of HCC44, H23, H1993, and HCC78 cells after 48 h treatment with the KRAS^G12C^ inhibitor AMG510 (4 μM). (**c**) EZH2 and MTHFD2 expression in AC cells after treatment with AMG510 (4 μM) for 48 h. (**d**) Western blot analysis of EZH2 and MTHFD2 expression in H1993 cells transfected with Kras^G12Cvec^ or Kras^WTvec^ plasmids. (**e**) Cellular survival of Kras^WTvec^- and Kras^G12Cvec^-transfected H1993 cells after 48 h treatment with AMG510 (4 μM). (**f**) Western blot analysis of indicated proteins in Kras^WTvec^- or Kras^G12Cvec^-transfected H1993 cells that were treated as in (**e**). (*** *p* < 0.001, **** *p* < 0.0001).

**Figure 3 metabolites-12-00652-f003:**
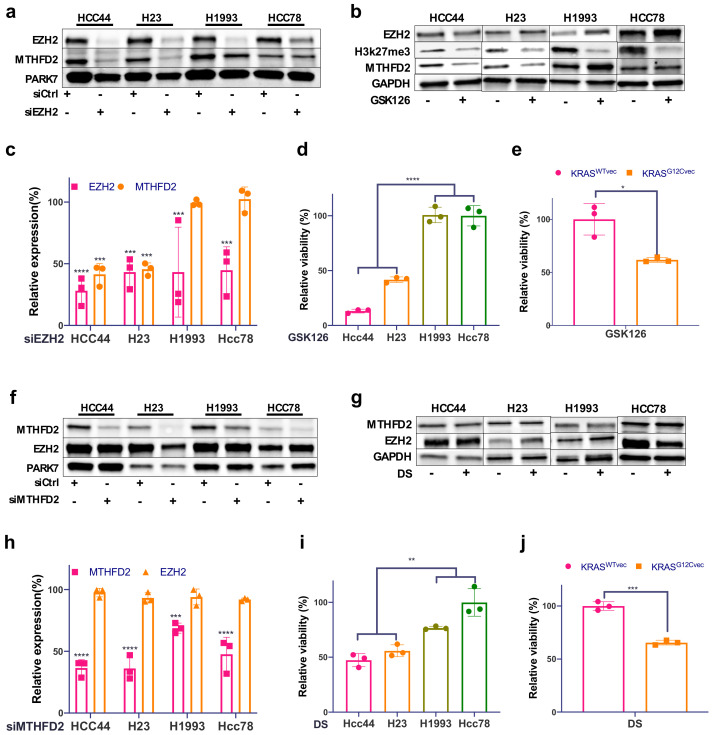
MTHFD2 expression is dependent on EZH2 in KRAS^G12C^ cell lines. (**a**) Western blot analysis of EZH2 and MTHFD2 in the four AC cell lines HCC44, H23, H1993, and HCC78 after EZH2 siRNA knockdown and (**b**) after 48 h treatment with the EZH2 inhibitor GSK126 (5 μM). (**c**) Related quantification of EZH2 and MTHFD2 protein expression after EZH2 knockdown. (**d**) The cellular survival of the four described cell lines treated as in (**b**). (**e**) The cellular survival of KRAS^WT^- and KRAS^G12C^-transfected H1993 cells after 48h treatment with GSK126 (5μM). (**f**) Western blot analysis of EZH2 and MTHFD2 in the four AC cell lines after MTHFD2 siRNA knockdown and (**g**) after 48 h treatment with the MTHFD2 inhibitor DS (20 μM). (**h**) Related quantification of EZH2 and MTHFD2 protein expression after MTHFD2 knockdown. (**i**) The cellular survival of the four described cell lines treated as in (**g**). (**j**) The cellular survival of KRAS^WTvec^ and KRAS^G12Cvec^ H1993 cells after 48 h treatment with DS (20 μM). (* *p* < 0.05, ** *p* < 0.01, *** *p* < 0.001, **** *p* < 0.0001).

**Figure 4 metabolites-12-00652-f004:**
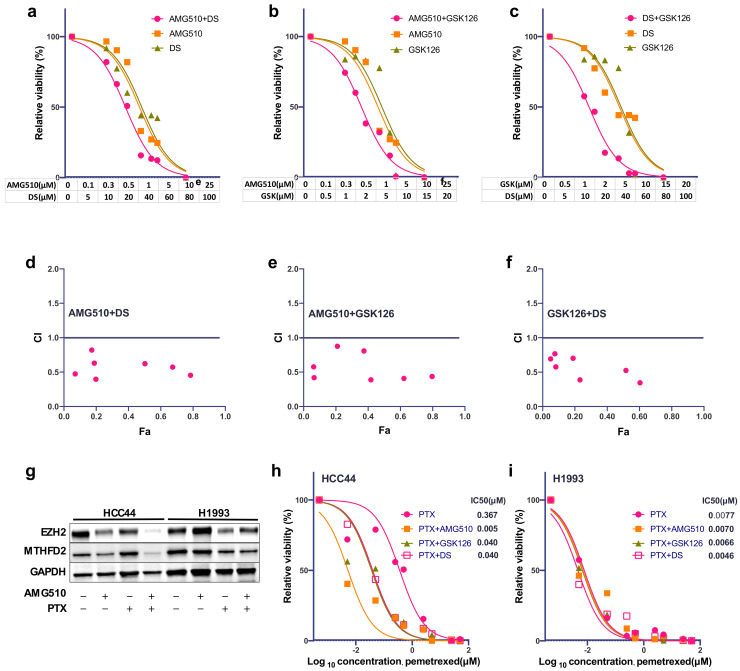
Combinational treatment of KRAS^G12C^ cells with EZH2 and MTHFD2 inhibitors. (**a**) Cellular survival of HCC44 after treatment with AMG510, DS, or the combined treatment at indicated concentrations for 72 h. (**b**) Cellular survival of HCC44 after treatment with AMG510, GSK126, or the combined treatment at indicated concentrations for 72 h. (**c**) Cellular survival of HCC44 after treatment with DS, GSK126, or the combined treatment at indicated dosages for 72h. (**d**–**f**) Combination index (CI) plot for AMG510, DS, and GSK126 in HCC44. CI values are plotted as a function of the fractional inhibition of cell viability by computer simulation using CompuSyn; the pink dots represent the actual experimental points. CI values: synergism (CI  <  0.9), additive effect (CI  =  0.9–1.1), and antagonism (CI  >  1.1). (**g**) Western blot analysis of EZH2 and MTHFD2 expression of HCC44 and H1993 after 48 h single treatment with AMG510 (2 μM) or PTX (20 μM) or in combination. (**h**,**i**) Dose–response curve of HCC44 and H1993 after 72 H single treatment with PTX (0.005–50 μM) alone or in combination with AMG510 (2 μM), GSK126 (2.5 μM), or DS (10 μM), respectively.

**Table 1 metabolites-12-00652-t001:** Clinical data summary.

Histology	Pulmonary Adenocarcinoma
Total	109
Median age (range)	67 (34–85)
Gender, n (%)	
Female	48 (44.0)
Male	61 (56.0)
Degree of differentiation, n (%)	
I + II	78 (71.6)
III	31 (28.4)
T-stage, n (%)	
I + II	90 (82.6)
III + IV	19 (17.4)
Lymph node metastasis, n (%)	
No	62 (60.2)
Yes	41 (39.8)
pUICC, n (%)	
I + II	82 (75.2)
III + IV	27 (24.8)
Median survival time (months)	23
Reported deaths (%)	52 (47.7)

**Table 2 metabolites-12-00652-t002:** Correlation of EZH2 expression and KRAS mutation with clinicopathologic parameters.

	IHC-EZH2	(n = 109)			KRAS (n = 62)			
Feature	Cases	−	+	*p*-Value	Cases	WT	MUT	*p*-Value
Gender, n (%)								
Female	48 (44)	33 (68.8)	15 (31.3)	0.0576	30 (48.4)	23 (76.7)	7 (23.3)	0.085
Male	61 (56)	34 (55.7)	27 (44.3)		32 (51.6)	21 (65.6)	11 (34.4)	
Age								
≥60	83 (76.1)	47 (56.6)	36 (43.4)	0.0026 **	46 (74.2)	35 (76.1)	11 (23.9)	0.0026 **
<60	26 (23.9)	20 (76.9)	6 (23.1)		16 (25.8)	9 (56.3)	7 (43.7)	
Degree of differentiation, n (%)								
G1–2	78 (71.6)	54 (69.2)	24 (30.8)	<0.0001 ***	47 (75.8)	35 (74.5)	12 (25.5)	0.035 *
G3	31 (28.4)	13 (41.9)	18 (58.1)		15 (24.2)	9 (60)	6 (40)	
T-stage, n (%)								
T1–2	90 (82.6)	55 (61.1)	35 (38.9)	0.7708	48 (77.4)	34 (70.8)	14 (29.2)	>0.99
T3–4	19 (17.4)	12 (63.2)	7 (36.8)		14 (22.6)	10 (71.4)	4 (28.6)	
Lymph node metastasis, n (%)								
No	62 (60.2)	44 (71.0)	18 (29)	0.0003 ***	43 (69.4)	29 (67.4)	14 (32.5)	0.0052 **
Yes	41 (39.8)	19 (46.3)	22 (53.7)		19 (30.6)	16 (84.2)	3 (15.8)	
pUICC, n (%)								
I + II	82 (75.2)	52 (63.4)	30 (36.6)	0.3133	51 (82.3)	36 (70.6)	15 (29.4)	0.7528
III + IV	27 (24.8)	15 (55.6)	12 (44.4)		11 (17.7)	8 (72.7)	3 (27.2)	

* *p* < 0.05, ** *p* < 0.01, *** *p* < 0.001.

## Data Availability

The data presented in this study are available in article and Appendix A.

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
