# Peer review of "Regulation and Therapeutic Targeting of MTHFD2 and EZH2 in KRAS-Mutated Human Pulmonary Adenocarcinoma"

_metabolites, 2022, doi:10.3390/metabo12070652_

Round 1
Reviewer 1 Report
The authors correlated mutated KRAS with MTHFD2 and EZH2 levels in tumor samples and confirmed the relationship in cell lines. They showed the increased efficacy of co-inhibition and efficacy of the specific KRASG12C inhibition. The presented data justify the conclusions made by the authors and there are only minor issues.
Minor
4.2: For completeness, mention also embedding media and deparaffinization. Indicate also, what the definition of weak and strong staining was (number or intensity of stained cells).
4.7: Indicate software for quantitative analysis of band intensity.
Author Response
- For completeness, mention also embedding media and deparaffinization. Indicate also, what the definition of weak and strong staining was (number or intensity of stained cells).
We would like to thank the reviewer for the critical reading, and we agree with the suggestions. We added the missing information in the material and methods part.
4.1. Human tissue samples
The NSCLC tissue samples used in this study were collected from the Department of Thoracic and Cardiovascular Surgery, University Medical Center Göttingen after surgical resections. Tissues were formalin-fixed in 4% buffered formaldehyde and paraffin-embedded for diagnostic purposes. The performed experiments were approved by the ethics committee of the University Medical Center Göttingen (#1-2-08, 24-4-20) and all procedures were performed in accordance with the seventh version of the Declaration of Helsinki [1].
4.2. Immunohistochemical staining
Tissue samples were assembled in tissue microarrays and EZH2 was immunohistochemically stained as described previously [2-4]. Briefly, 2µm tissue sections were deparaffinized, rehydrated and subsequently incubated in EnVision Flex Target Retrieval Solution pH high (Dako, Japan) followed by incubation with primary antibody against EZH2 (Leica, Germany, NCL-L-EZH2, 1:50), secondary antibody (EnVision Flex+, Dako), DAB (Dako) and counterstained with Hematoxylin. Staining was evaluated under light microscopy according to signal intensity of the stained cells: zero = no staining; one = weak staining intensity; two = strong staining intensity.
- Indicate software for quantitative analysis of band intensity.
We added the software used for the quantification of the WB signals in the material and methods parts as follows.
4.9. Statistical analysis
Statistical analysis was performed using GraphPad (GraphPad Software LCC). Two-group comparisons were performed using students' t-tests. Half maximal inhibitory concentration (IC50) was performed by Pearson’s correlation test. Cell growth and resistance comparison were analyzed using two-way ANOVA. Survival analyses were performed using the Kaplan-Meier method and Log-Rank (Cox-Mantel) test. ImageJ was used for the quantification of WB signal intensities [5]. Three biological replicates were performed. The data depicted are the mean ± SEM. A p-value of < 0.05 was considered significant (* p < 0.05, ** p < 0.01, *** p < 0.001).
Reviewer 2 Report
In the experimental study the enzymes methylenetetrahydrofolate dehydrogenase 2 (MTHFD2) and enhancer of Zeste Homolog 2 (EZH2) were addressed as potential therapeutic targets in KRAS mutated non-small cell lung cancer (NSCLC), in particular pulmonary adenocarcinomas. Using human tissues as well as established cell lines (HCC44, H23, H1993, and HCC78) the authors demonstrate data indicating that mutated KRAS is associated with an increased susceptibility of tumor cells to MTHFD2 and EZH2 inhibitors.
The ms is well-written and illustrated. The topic is in the scope of the Journal.
Comments
1. Abstract: MTHFD2 and EZH2 should be given in full-length.
2. Introduction: The one-carbon metabolism pathway should be introduced to the reader with a short paragraph.
3. Material and Methods: The molecular profiling of the human tumor tissues comes not clear to the reader. The G12C is exclusive? – or is there any other mutation.
4. Results: The methylation status of cultured cells could be of interest to elucidate the molecular link in a semi-functional way. Is there any data available?
5. Discussion: a putative molecular pathway should be suggested. The statement that further studies will be necessary to elucidate the molecular mechanisms is poor.
